# Enhanced Carbon Dioxide Decomposition Using Activated SrFeO$_{3-\delta}$

**Jaeyong Sim [1,2], Sang-Hyeok Kim [1,2], Jin-Yong Kim [1,2], Ki Bong Lee [2,*], Sung-Chan Nam [1] and Chan Young Park [1,*]**

[1] Greenhouse Gas Research Laboratory, Korea Institute of Energy Research, 152 Gajeong-ro, Yuseong, Daejeon 34129, Korea; resumemat@naver.com (J.S.); ksh0630ksh@kier.re.kr (S.-H.K.); tyui15678@kier.re.kr (J.-Y.K.); scnam@kier.re.kr (S.-C.N.)

[2] Department of Chemical and Biological Engineering, Korea University, 145 Anam-ro, Seongbuk-gu, Seoul 02841, Korea

\* Correspondence: kibonglee@korea.ac.kr (K.B.L.); cpark@kier.re.kr (C.Y.P.); Tel.: +82-2-3290-4851 (K.B.L.); +82-42-860-3069 (C.Y.P.)

**Abstract:** Today, climate change caused by global warming has become a worldwide problem with increasing greenhouse gas (GHG) emissions. Carbon capture and storage technologies have been developed to capture carbon dioxide ($CO_2$); however, $CO_2$ storage and utilization technologies are relatively less developed. In this light, we have reported efficient $CO_2$ decomposition results using a nonperovskite metal oxide, $SrFeCo_{0.5}O_x$, in a continuous-flow system. In this study, we report enhanced efficiency, reliability under isothermal conditions, and catalytic reproducibility through cyclic tests using $SrFeO_{3-\delta}$. This ferrite needs an activation process, and 3.5 vol% $H_2/N_2$ was used in this experiment. Activated oxygen-deficient $SrFeO_{3-\delta}$ can decompose $CO_2$ into carbon monoxide (CO) and carbon (C). Although $SrFeO_{3-\delta}$ is a well-known material in different fields, no studies have reported its use in $CO_2$ decomposition applications. The efficiency of $CO_2$ decomposition using $SrFeO_{3-\delta}$ reached ≥90%, and decomposition (≥80%) lasted for approximately 170 min. We also describe isothermal and cyclic experimental data for realizing commercial applications. We expect that these results will contribute to the mitigation of GHG emissions.

**Keywords:** greenhouse gas; climate change; $CO_2$ decomposition; $CO_2$ utilization; $SrFeO_{3-x}$

## 1. Introduction

Climate change caused by global warming has emerged as a problem worldwide owing to the increase in greenhouse gas (GHG) emissions. Carbon dioxide ($CO_2$) emissions account for more than 90% of global GHG emissions [1,2]. According to the Intergovernmental Panel on Climate Change report in 2018, $CO_2$ emissions were estimated to be at 32 and 40 billion tons in 2010 and 2020, respectively [3]. Carbon capture and storage technologies have been developed to capture $CO_2$ emissions, especially those from power plants [4–6]. However, $CO_2$ storage is vulnerable to earthquakes and can cause pollution and is, therefore, recognized as only a temporary method [7]. Therefore, there is an urgent need to develop and implement techniques for utilizing captured $CO_2$. For example, $CO_2$ reforming of methane has been proposed; however, it has high cost and energy requirements [8,9].

One of the treatment methods for captured $CO_2$ is catalytic decomposition using oxygen-deficient metal oxides. Tamauara et al. reported that oxygen-deficient magnetite ($Fe_{3+\delta}O_4$, $\delta = 0.127$) decomposed up to 100% of $CO_2$ and $H_2O$ at 290 °C [10]. Subsequently, $CO_2$ decomposition using ferrites with divalent metals such as $Ni^{2+}$ and $Cu^{2+}$ was investigated [11]. $Mn^{2+}$- and $Zn^{2+}$-based ferrites were also

reported as having $CO_2$ decomposition efficiencies of 66% and 90%, respectively [12,13]. Even trivalent, Ni-Cu ferrites were tested for an identical purpose [14]. It was reported that nickel and copper substitutions at the A-site of ferrites were the most beneficial for reduction and oxidation reactions and demonstrated meaningful results. However, all these approaches are difficult to apply in practical applications because the experimental data were obtained from a small stagnant batch-type reactor.

The first attempt to go beyond the batch system was made in 2001. Shin et al. reported $CO_2$ decomposition data obtained through thermogravimetric analysis using activated $CuFe_2O_4$ [11]. Furthermore, Kim et al. investigated $CO_2$ decomposition using activated $Ni_{0.5}Zn_{0.5}Fe_2O_{4-\delta}$ in a continuous flow of 10% $CO_2$-balanced $N_2$ [15,16]. They reported that trivalent ferrites (i.e., $(Ni_xZn_{1-x})Fe_2O_4$, x = 0.3, 0.5, 0.7, and 1) showed a higher $CO_2$ decomposition efficiency than divalent $NiFe_2O_4$ ferrite. They ascertained that the ferrites could completely decompose 10% $CO_2$ for 5 to 7 min. They also asserted that Ni/Zn-ferrite synthesized by the hydrothermal method displayed better $CO_2$ decomposition performance than that synthesized by the coprecipitation method. However, they did not perform a blank test and quantitative analysis. Although their results were elementary and had some weaknesses, their trials were invaluable in that they can be applied in practical applications. Therefore, the accumulated data of $CO_2$ decomposition in a continuous system should be obtained for realizing economically efficient $CO_2$ treatment.

In our previous work [17], we demonstrated the possibility of continuous $CO_2$ decomposition by using oxygen-deficient metal oxides and suggested its reaction mechanism. Compared to Ni-ferrites, a nonperovskite-type metal oxide (i.e., $SrFeCo_{0.5}O_x$) was much more effective for $CO_2$ decomposition: Ni-ferrites decomposed only up to 20% of $CO_2$, whereas $SrFeCo_{0.5}O_x$ displayed a $CO_2$ decomposition efficiency of up to 90%. These results were obtained based on our suggested mechanism that high electrical and ionic conductivities affect $CO_2$ decomposition. Currently, the obtainment of suitable isothermal and regeneration data will be more helpful for practical applications. In our ongoing research project, we have found that another material, $SrFeO_{3-\delta}$, shows greater promise for this purpose.

Originally, $SrFeO_{3-\delta}$ was used as an oxygen transport material [18–23] and as a catalyst for methane combustion and chemical looping processes [24,25]. Perovskite-type $SrFeO_{3-\delta}$ ($0 \leq \delta \leq 0.5$) is a nonstoichiometric metal oxide containing Fe ions in a mixed valence, such as $Fe^{4+}$ and $Fe^{3+}$ [26]. Under reducing conditions, $SrFeO_{3-\delta}$ produces oxygen vacancies; the number of oxygen vacancies depends on the temperature and the oxygen partial pressure [27]. Recently, Marek et al. reported the stable use of $SrFeO_{3-\delta}$ in chemical looping systems; the material reduced above $\delta = 0.5$ could be reoxidized with either $CO_2$ or air, resulting in $SrFeO_{3-\delta}$ ($0 \leq \delta \leq 0.5$) [25]. Therefore, we consider it a promising material for $CO_2$ decomposition. Several studies have reported on the use of $SrFeO_{3-\delta}$ in various fields. However, no study has reported the use of $SrFeO_{3-\delta}$ for $CO_2$ decomposition in a continuous-flow system. In this report, we describe the reduction behavior and redox reaction of $SrFeO_{3-\delta}$. Furthermore, through cyclic experiments, we demonstrate that it exhibits consistently high $CO_2$ decomposition performance under isothermal conditions. We also demonstrate its structural stability as a catalytic material for practical applications.

## 2. Results

### 2.1. Characterization

The crystal structure of $SrFeO_{3-\delta}$ was analyzed by X-ray powder diffraction (XRD) at 40 kV and 200 mA. The XRD powder patterns of the samples were obtained in 0.02° steps over the range of $20° \leq 2\theta \leq 80°$. It has been reported that the structure of $SrFeO_{3-\delta}$ could be changed susceptibly by $\delta$ values, namely, cubic at $\delta = 0–0.12$, tetragonal at $\delta = 0.16–0.24$, and orthorhombic at $\delta = 0.25$ [28,29]. The lattice constant of $SrFeO_{3-\delta}$ obtained from XRD data was a = 5.479(5) Å, b = 7.729(8) Å, c = 5.521(2) Å, and V = 233.8(5) Å$^3$. It was determined to be an orthorhombic perovskite, which is in good agreement with the reported value (PDF# 01-077-9154). Figure 1a shows XRD powder patterns of as-synthesized $SrFeO_{3-\delta}$. We also obtained secondary electron images. These are discussed with those measured

after $CO_2$ decomposition tests at the end of this section. The chemical composition was reasonably acceptable, and the surface area of $SrFeO_{3-\delta}$ was determined to be 3.19 $m^2/g$.

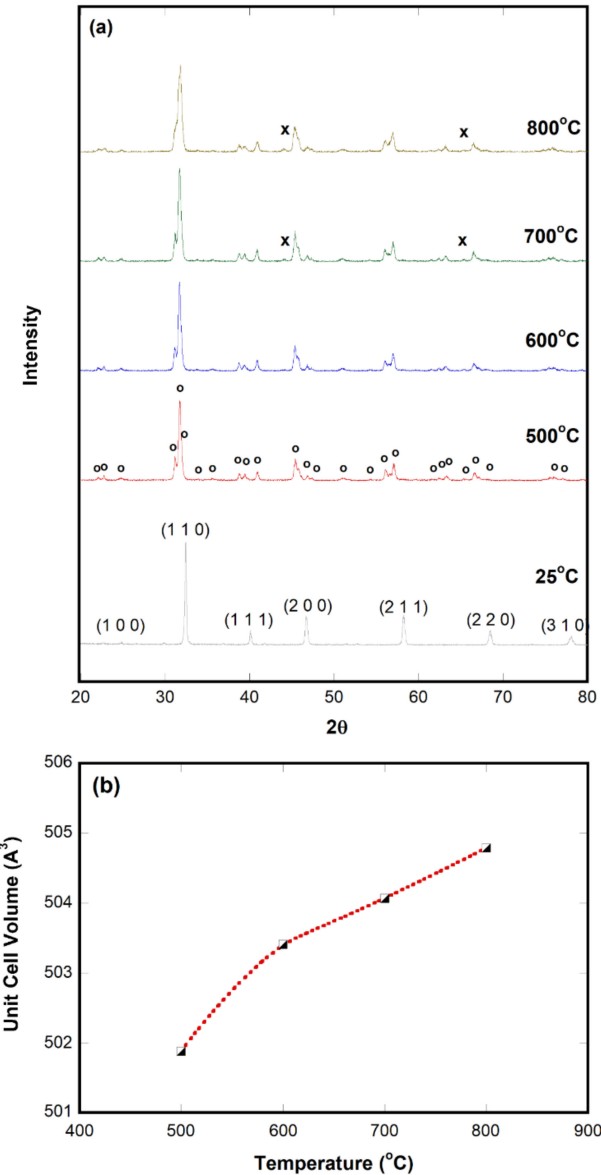

**Figure 1.** In-situ XRD results of $SrFeO_{3-\delta}$: (**a**) In-situ XRD powder pattern and (**b**) unit cell volume at $500 \leq T \leq 800$ °C. The symbols indicate Fe metal (x) and brownmillerite (o).

## 2.2. Oxygen-Deficient $SrFeO_{3-\delta}$

As discussed in our previous report [17], sample activation is an essential step to decompose $CO_2$. An understanding of the reduction behavior is also needed because $CO_2$ decomposition could mainly be affected by the number of oxygen vacancies and their mobility. $CO_2$ decomposition is induced by the incorporation of $O^{2-}$ into oxygen vacancies. In-situ XRD was performed during high-temperature reduction with 3.5 vol% $H_2/Ar$ to identify structural changes occurring in the metal oxide. Figure 1 includes the in-situ X-ray powder patterns of $SrFeO_{3-\delta}$ obtained at $500 \leq T \leq 800$ °C.

$SrFeO_{3-\delta}$ was observed in the perovskite phase at room temperature, and the pattern was very similar to that of $SrFeO_{2.75}$ (PDF# 01-077-9154). As the temperature increased, the perovskite would lose more oxygen and change to a brownmillerite phase. Phase changes from $SrFeO_{3-\delta}$ to $SrFeO_{2.5}$ at 500 °C are attributed to the partial reduction of $Fe^{4+}$ to $Fe^{3+}$ [30]. An almost pure brownmillerite phase

(SrFeO$_{2.5}$, PDF# 01-070-0836) was observed at 500–600 °C. The XRD pattern at 500 °C was completely indexed with an orthorhombic unit cell with lattice parameter a = 5.69(9) Å, b = 15.80(2) Å, c = 5.57(2) Å, and V = 501.8(0) Å$^3$. This indicated that the cell volume increased by more than twice via the expansion of one side in the orthorhombic unit cell, especially the b-axis. Sr$_{n+1}$Fe$_n$O$_{3n+1}$ and Fe$^0$ peaks have been reported to appear when SrFeO$_{2.5}$ was reduced further by increasing temperature and reaction time [25]. SrO and Fe$^0$ peaks are considered the final products of the reduction. In our patterns, only a trace of Fe metal (PDF# 01-080-3817) peaks appeared at 2θ ≈ 44.1° and 65.4° at ≥700 °C. Typical Fe$^0$ peaks could be observed at 2θ ≈ 44.0° and 65.3°. The Sr$_3$Fe$_2$O$_{6.14}$ phase might be possible; however, it overlaps with brownmillerite peaks. In addition, based on our thermogravimetric result (not shown), the change in the nonstoichiometric value (δ) was determined to be ~0.8 at 25 °C ≤ T ≤ 800 °C. Figure 1b shows the calculated unit cell volumes as a function of temperature. They demonstrate linearity over 600 °C [31]. It has been noted that SrFeO$_{3-\delta}$ should be activated without complete structural collapse. If these phase changes are reversible, it would be beneficial for catalyst redox reactions or in a chemical looping system.

### 2.3. Effect of Conductivity on CO$_2$ Decomposition

As we suggested a mechanism in our previous report, CO$_2$ decomposition is considered to be affected significantly by the conductivity of the sample [17]. To decompose CO$_2$ effectively, the metals in the ferrite should easily provide electrons for CO$_2$; therefore, the electronic conductivity plays an important role at first. As the oxygen in neutral CO$_2$ has sufficient electrons, electron transfer from metals on the surface of the ferrite to CO$_2$ is not likely to occur naturally. Therefore, the activation process should be performed before exposure to CO$_2$, and the produced oxygen vacancies become the driving force of the redox reaction. Based on such reasoning, the amount of oxygen vacancies could be the most important factor in this reaction. In addition, the activation process would be easier if the metal oxide contained metals with variable oxidation number.

Once the oxygen ions (i.e., O$^{2-}$) fill the oxygen vacant sites on the sample surface, the ability to decompose CO$_2$ to CO or C is lost owing to the saturation (or deactivation) of the sample surface. However, if oxygen ions migrate well inside the lattice and oxygen vacancies are reformed on the sample surface, CO$_2$ decomposition could be continued until all vacancies are filled. This is why oxygen ionic conductivity should also be considered. It would be interesting to determine how much of a role oxygen ion conductivity plays in the decomposition of carbon dioxide; however, this will be reported in a separate paper. After all, the oxygen ions accepted through the electronic conductivity effect could move to inside defects via oxygen ionic conducting properties. Therefore, samples with good electrical and ionic conductivity would decompose CO$_2$ more effectively. The total conductivity of SrFeO$_{3-\delta}$ is 31.6 S cm$^{-1}$ at 800 °C [32], which is higher than that of SrFeCo$_{0.5}$O$_x$ (17 S cm$^{-1}$) [33]. The total conductivity of SrFeO$_{3-\delta}$ shows good agreement with the reference value, and it was determined to be 33.9 S cm$^{-1}$ at 800 °C from our own measurement. This feature was the reason that SrFeO$_{3-\delta}$ was selected for the CO$_2$ decomposition experiment in this paper.

### 2.4. CO$_2$ Decomposition

To the best of our knowledge, only two studies have reported CO$_2$ decomposition in a continuous gas-flow reactor before our previous report [17]. One [11] provided TGA measurement results, and the other [15] provided extremely limited information. We have used several metal oxides for CO$_2$ decomposition experiments in a continuous-flow reactor. In our previous report, we reported CO$_2$ decomposition with SrFeCo$_{0.5}$O$_x$ using data obtained under nonisothermal conditions. Even if nonisothermal data are insufficient to cover all practical applications, they can serve as a cornerstone to determine the most economically efficient temperature region for CO$_2$ decomposition. Furthermore, temperature fluctuates during both activation and decomposition processes. Considering these applications, we performed nonisothermal tests, isothermal tests, and cyclic experiments for CO$_2$

decomposition with a noncobalt metal oxide, $SrFeO_{3-\delta}$. A comparison of nonisothermal data for $SrFeCo_{0.5}O_x$ and $SrFeO_{3-\delta}$ is presented below.

Nonisothermal $CO_2$ decomposition: Figure 2 shows a comparison of the results of nonisothermal $CO_2$ decomposition using $SrFeO_{3-\delta}$ and $SrFeCo_{0.5}O_x$ for temperatures ranging between 25 and 800 °C. Data for $SrFeCo_{0.5}O_x$ were extracted from our previous report [17], and the same experimental conditions were applied. Initially, we started $CO_2$ decomposition with $NiFe_2O_4$ as an oxygen-deficient ferrite; however, it decomposed only up to 20% of $CO_2$ in the continuous gas-flow system. We obtained a ~90% efficiency of $CO_2$ decomposition using $SrFeCo_{0.5}O_x$ selected based on our proposed mechanism. Here, we demonstrated several enhanced $CO_2$ decomposition results obtained by using $SrFeO_{3-\delta}$. First, $SrFeO_{3-\delta}$ could be activated at a much lower temperature and for shorter duration. $SrFeO_{3-\delta}$ was primarily activated at 280 ≤ T ≤ 600 °C, as shown in Figure 2a. The $H_2$ concentration during reduction decreased rapidly up to ≈460 °C, which indicated the phase changes from perovskite to brownmillerite. This behavior can be confirmed from the in-situ XRD data shown in Figure 1. It is believed that oxygen vacancies are created the most at these temperatures. Second, the amount of $CO_2$ decomposed using $SrFeO_{3-\delta}$ is approximately 2.2 times higher than that decomposed using $SrFeCo_{0.5}O_x$ based on the calculated result of ≥50% $CO_2$ decomposition, as shown in Figure 2b. As the temperature increased, the $CO_2$ decomposition efficiencies of both metal oxides increased by up to ~90%. After reaching 800 °C, the $CO_2$ decomposition efficiency of $SrFeCo_{0.5}O_x$ decreased, whereas the high decomposition efficiency of $SrFeO_{3-\delta}$ was maintained over 100 min. This indicates that $SrFeO_{3-\delta}$ might be a more appropriate material for $CO_2$ decomposition than $SrFeCo_{0.5}O_x$. The amount of CO produced using $SrFeO_{3-\delta}$ was also slightly higher than that produced using $SrFeCo_{0.5}O_x$.

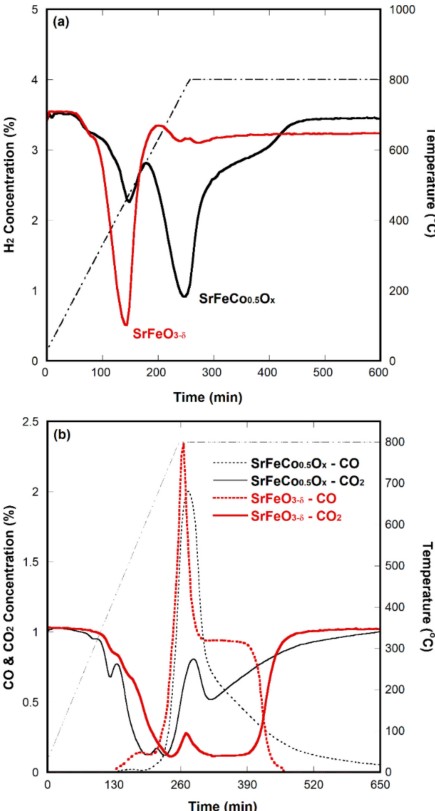

**Figure 2.** $CO_2$ decomposition results: (**a**) Consumed $H_2$ concentration and (**b**) $CO_2$ and CO concentrations during sample activation and $CO_2$ decomposition tests, respectively. The black lines indicate the results of $SrFeCo_{0.5}O_x$ extracted from our previous work [17], and the straight dotted lines indicate temperature profiles (i.e., 3 °C/min).

The $CO_2$ decomposition ability could be expressed in units of millimoles of decomposed $CO_2$ and generated CO per gram of sample loaded (i.e., mmol $g^{-1}$). This calculation was made using several assumptions. For example, the final decomposition was determined to be the point at which $CO_2$ decomposition ceased, resulting in the revelation of the initial $CO_2$ concentration. The final $CO_2$ decomposition time was determined during the point at which $SrFeO_{3-\delta}$ started to decompose at 200 °C and continued for over 4 h, even reaching 800 °C. This calculation was made using the decomposition time limit that ranged between 54 and 500 min (see Figure 2b). Secondly, in spite of nonisothermal $CO_2$ decomposition, the ideal gas law was used to calculate the decomposed amount (i.e., mmol $g^{-1}$) of input $CO_2$. The exact same conditions were applied for the $NiFe_2O_{3-\delta}$ and $SrFeCo_{0.5}O_x$ samples. The results are summarized in Table 1. Both $CO_2$ decomposition and CO generation using a perovskite ($SrFeO_{3-\delta}$) demonstrated enhanced performance compared to those using a spinel ($NiFe_2O_{3-\delta}$) or a nonperovskite ($SrFeCo_{0.5}O_x$). In addition, $SrFeO_{3-\delta}$ is a cobalt-free compound that is economical and environmentally friendly. Generally, cobalt-containing metal oxides display good catalytic behavior but have several shortcomings, such as structural instability even at intermediate operating temperatures (500 to 800 °C) in a long-term test [34]. In the case of $NiFe_2O_{3-\delta}$, other shortcomings were observed, such as too long an oxidation time.

**Table 1.** The rate of decomposed $CO_2$ and generated CO in nonisothermal experiments.

| Sample | Decomposed $CO_2$ (mmol/g) | Produced CO (mmol/g) | Reference |
|---|---|---|---|
| $NiFe_2O_{4-\delta}$ | 2.25 | 2.89 | [17] |
| $SrFeCo_{0.5}O_x$ | 2.35 | 2.65 | [17] |
| $SrFeO_{3-\delta}$ | 3.30 | 2.95 | This work |

Isothermal $CO_2$ decomposition: For practical applications, data for $CO_2$ decomposition using $SrFeO_{3-\delta}$ at constant temperature should be determined. Figure 3 shows the isothermal results. The measurements were performed at 500, 600, 625, 650, 700, and 800 °C. The temperatures for sample activation and decomposition were identically controlled, and a blank test was also performed in the same reactor. GC was used to determine data points every ~4 min after switching the gas with 1 vol% $CO_2$/He. Fresh powder samples were used in each measurement. As the temperature increased, the amounts of $CO_2$ decomposition and CO production increased (see Figure 3a,b). This is probably attributable to the amount and high mobility of oxygen vacancies at higher temperatures. The ionic conductivities are proportional to the mobility of perovskite metal oxides (i.e., $\sigma = n \times e \times \mu$, where $\sigma$ is the specific conductivity; n, the number of charge carriers of a species; e, its charge; and $\mu$, its mobility) and generally increases with the temperature [35].

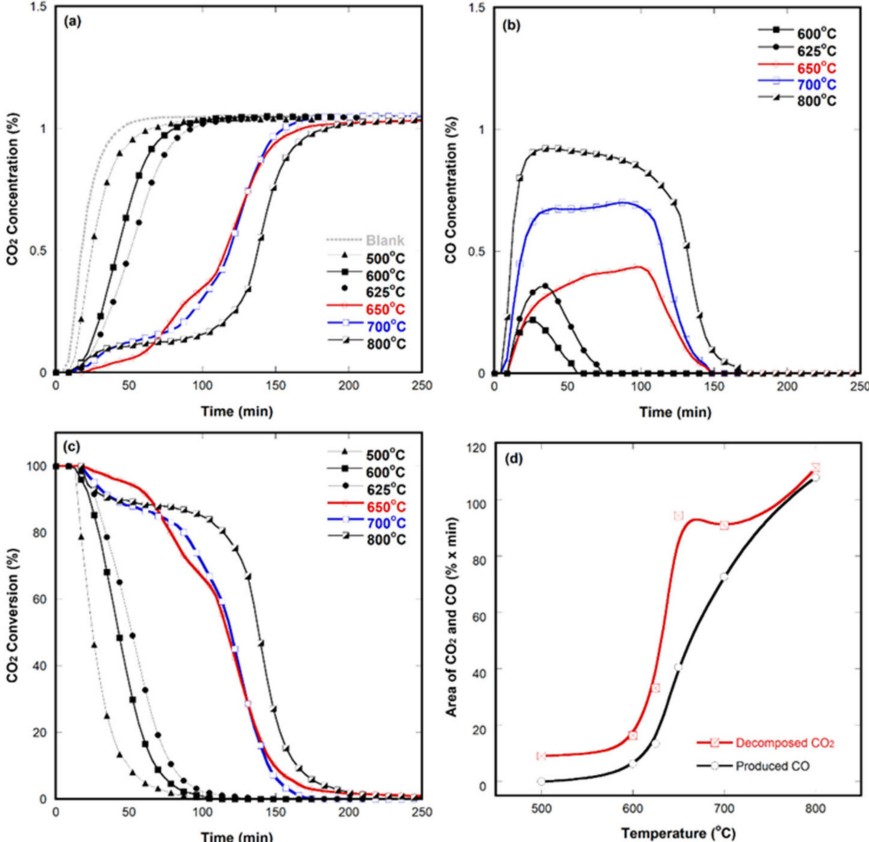

**Figure 3.** Isothermal $CO_2$ decomposition results obtained using $SrFeO_{3-\delta}$ at various temperatures: (**a**) Decomposed $CO_2$ concentration, (**b**) produced CO concentration, (**c**) $CO_2$ conversion as a function of time, and (**d**) decomposed $CO_2$ and produced CO area extracted from (**a**) and (**b**).

The $CO_2$ decomposition results between 600 and 700 °C were noteworthy. As the operating temperature increased from 625 to 650 °C, the amount of $CO_2$ decomposition doubled. We performed in-situ XRD and TGA experiments to analyze this unusual behavior in this temperature region (not shown). However, no special structural phase or weight changes were seen in the sample activation process. The amounts of consumed hydrogen for sample activation and cell parameters also demonstrated no considerable difference. The reason for the sudden increase in $CO_2$ decomposition upon increasing temperature by only 25 °C remains unclear. We presume that the thermal energy at 650 °C might boost $CO_2$ decomposition and the reverse Boudouard reaction (i.e., C(s) + $CO_2$ → 2CO). The mobility increase caused by the thermal energy might be an important factor because other factors, such as the unit cell volume, oxygen ion vacancy concentration, and weight change from TGA, did not change abruptly. These issues are discussed further by comparing sample characteristics before and after performing measurements in Section 2.5.

Based on the obtained data, $CO_2$ conversion rates were calculated using Equation (1).

$$CO_2 \text{ Conversion } (\%) = \frac{CO_2 \text{ In } - CO_2 \text{ Out}}{CO_2 \text{ In}} \times 100 \tag{1}$$

Although Figure 3c illustrates the same data in the same format as Figure 3a, the conversion degree is easier to distinguish from the $CO_2$ conversion plot. Furthermore, ≥90% of $CO_2$ conversion lasted for ≈65 min at 650 °C. This drastic change was much more evident from the area plots of the decomposed $CO_2$ and produced CO shown in Figure 3d. We calculated these areas by subtracting those obtained in the isothermal blank tests. It should be noted that the area for $CO_2$ (i.e., amount of $CO_2$ decomposition) was unusually high at 650 °C. It was even slightly higher than that at 700 °C. Further, CO production

increased rapidly until the temperature was increased up to 800 °C. The shape of the isothermal $CO_2$ decomposition curve at 650 °C also slightly differed from those of the others. We plan to analyze these behaviors using temperature-programed reduction and temperature-programed oxidation in a separate paper.

### 2.5. Stability Tests

Irrespective of how high the efficiency is, samples should have reproducibility and long-term stability, especially in severe redox reactions. This is the most important criterion for practical applications. In this report, we performed $CO_2$ decomposition in five reproducibility tests. We also performed cyclic $CO_2$ decomposition tests with partially activated $SrFeO_{3-\delta}$; however, we did not include the data, owing to their overlapped content. After these stability tests, the changes in the structural behavior and surface morphology of the used $SrFeO_{3-\delta}$ were investigated.

Figure 4 shows the results of the five cyclic reproducibility tests for $CO_2$ decomposition using $SrFeO_{3-\delta}$ at 700 °C. The sample was activated for 800 min in each cycle. The $CO_2$ decomposition of each cycle was subtracted from the amount of the blank test. The measurement data of the last cycle (i.e., the fifth cycle) did not perfectly match those of the first. However, they were reasonably close, and the difference could be attributed to the annealing effect of lasting temperatures. Average amounts of decomposed $CO_2$ and generated CO were determined to be 1.38 and 1.02 mmol per gram of catalyst, respectively. This result indicated a certain level of coke generation or possible adsorption of part of the carbon dioxide instead of decomposition. Although $SrFeO_{3-\delta}$ demonstrated good reproducibility even after repeating the redox experiment several times, it still took too long to activate the sample. In these days, we are using coke oven gas from the steel industry for sample activation. As it contains 55% to 60% hydrogen, the activation time will be much faster. In addition, $SrFeO_{3-\delta}$ impregnated with a small amount of precious metals such as Ru and Rh is tested for low-temperature $CO_2$ decomposition. Table 2 summarizes the test results.

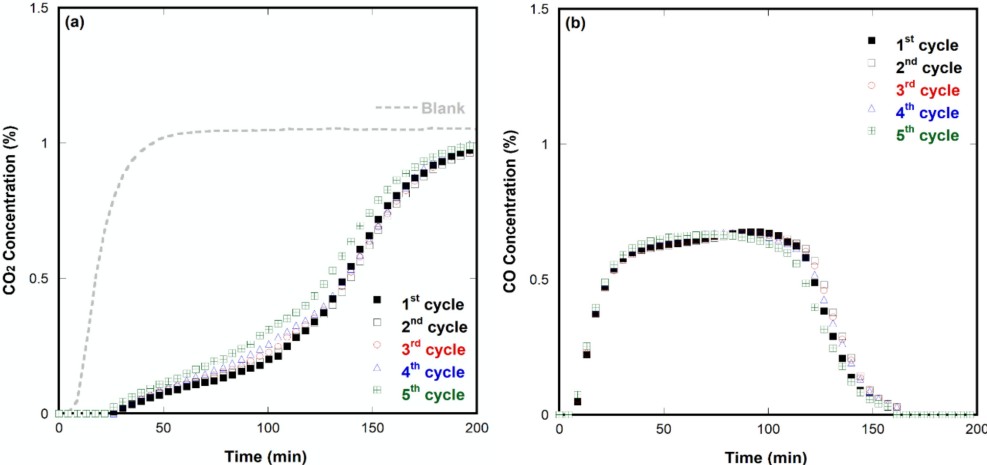

**Figure 4.** Five cyclic reproducibility tests for $CO_2$ decomposition using $SrFeO_{3-\delta}$ at 700 °C: (**a**) Decomposed $CO_2$ concentration and (**b**) produced CO concentration.

**Table 2.** Results of isothermal cyclic experiments with $SrFeO_{3-\delta}$.

| Temperature (°C) | Cycle (Number) | Decomposed $CO_2$ (mmol/g) | Produced CO (mmol/g) | Cell Parameters (Å) |
|---|---|---|---|---|
| | 1 | 1.40 | 1.00 | |
| | 2 | 1.40 | 1.03 | a = 5.66(8) |
| 700 | 3 | 1.39 | 1.03 | b = 15.59(2) |
| | 4 | 1.42 | 1.04 | c = 5.53(4) |
| | 5 | 1.29 | 0.98 | V = 489.1 Å$^3$ |
| | Average | 1.38 | 1.02 | |

To investigate the structural changes, XRD measurements were performed with the tested SrFeO$_{3-\delta}$ powders. Figure 5a shows the sample used for the nonisothermal CO$_2$ decomposition experiment at up to 800 °C. Figure 5b,d illustrate the powder patterns of SrFeO$_{3-\delta}$ that underwent cyclic tests at 700 and 650 °C, respectively. The brownmillerite phase remained, and it was hard to find impurities, including even traces of Fe-metal peaks. This indicates that the redox reaction is reversible and that SrFeO$_{3-\delta}$ can presumably serve as an excellent reactant. For comparison, the in-situ XRD result extracted from Figure 1 was added in Figure 5c. Three oxidized XRD patterns (i.e., Figure 5a,b,d) shifted to a higher 2θ angle; Figure 5c shows the pattern of the reduced sample. When the sample reduced, the oxygen vacancy concentration increased, resulting in its increased volume. The unit cell volume of reduced SrFeO$_{3-\delta}$ at 700 °C was 504.1 Å$^3$, and that of the oxidized sample decreased to 489.1 Å$^3$. Table 3 lists the fully indexed unit cell parameters for SrFeO$_{3-\delta}$ tested at 700 °C.

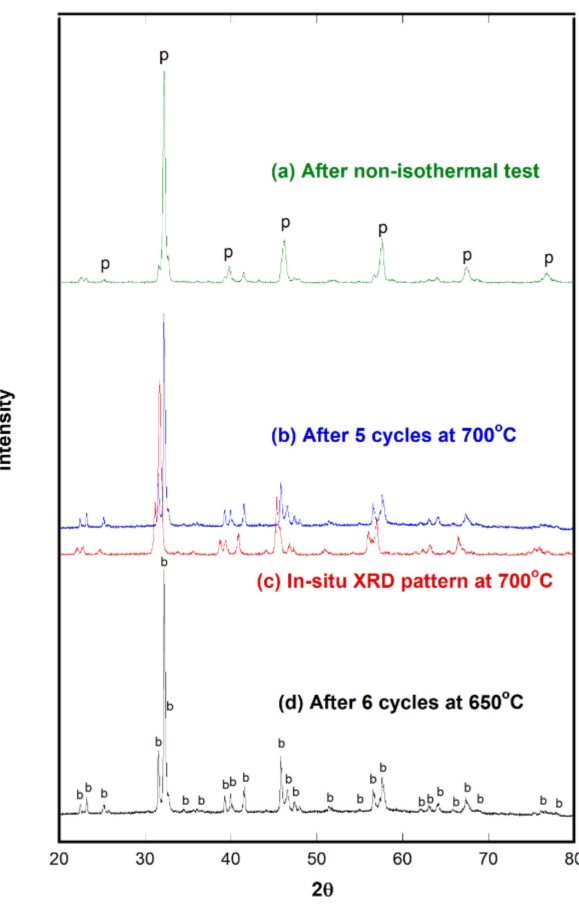

**Figure 5.** XRD powder patterns of SrFeO$_{3-\delta}$ tested for CO$_2$ decomposition measurements: (**a**) After nonisothermal test, (**b**) after five cycles at 700 °C, (**c**) in-situ XRD at 700 °C, and (**d**) after six cycles at 650 °C. The symbols indicate perovskite (p) and brownmillerite (b).

**Table 3.** Experimental conditions of nonisothermal and isothermal tests with SrFeO$_{3-\delta}$.

| Type | Temp. (°C) | Source | Conditions |
|---|---|---|---|
| Nonisothermal | 25–800 | Figure 2 | |
| Isothermal | 500 600 625 650 700 800 | Figure 3 | Activation: 3.5 vol% H$_2$/N$_2$ CO$_2$ decomposition: 1 vol% CO$_2$/He Flow rate: 50 mL/min Ramp rate: 3 °C/min |

It should be noted that the powder pattern of the sample tested at 800 °C shows a mixed phase, that is, perovskite and brownmillerite, whereas the sample tested at ≤700 °C primarily shows a brownmillerite phase. This result is relevant to the increased amounts of decomposed $CO_2$ at 800 °C (see Figure 3a). For a perovskite phase to exist at 800 °C, more oxygen vacancies need to be filled. This condition is induced by $CO_2$ decomposition. It should also be noted that three oxidized samples were slightly reduced with $N_2$ because 1 vol% $CO_2$/He gas was switched with $N_2$ after decomposition tests when cooling down to room temperature.

The microstructure of $SrFeO_{3-\delta}$ was examined using SEM before and after $CO_2$ decomposition experiments. Figure 6a shows a secondary electron image of pristine $SrFeO_{3-\delta}$, indicating that many small-sized (≤100 nm) particles are attached and dispersed on bigger ones. These small particles grew twice as large after the redox tests at 650 °C, and some large particles appeared to aggregate to each other (see Figure 6b). Even larger agglomerates developed and were observed in the sample tested at 700 °C, as shown in Figure 6c. Furthermore, the agglomerated particles (even ≥1 μm) displayed distinct grain boundaries. As the activity of samples is generally believed to depend on their surface area, a detailed microstructural investigation will be conducted in a separate study.

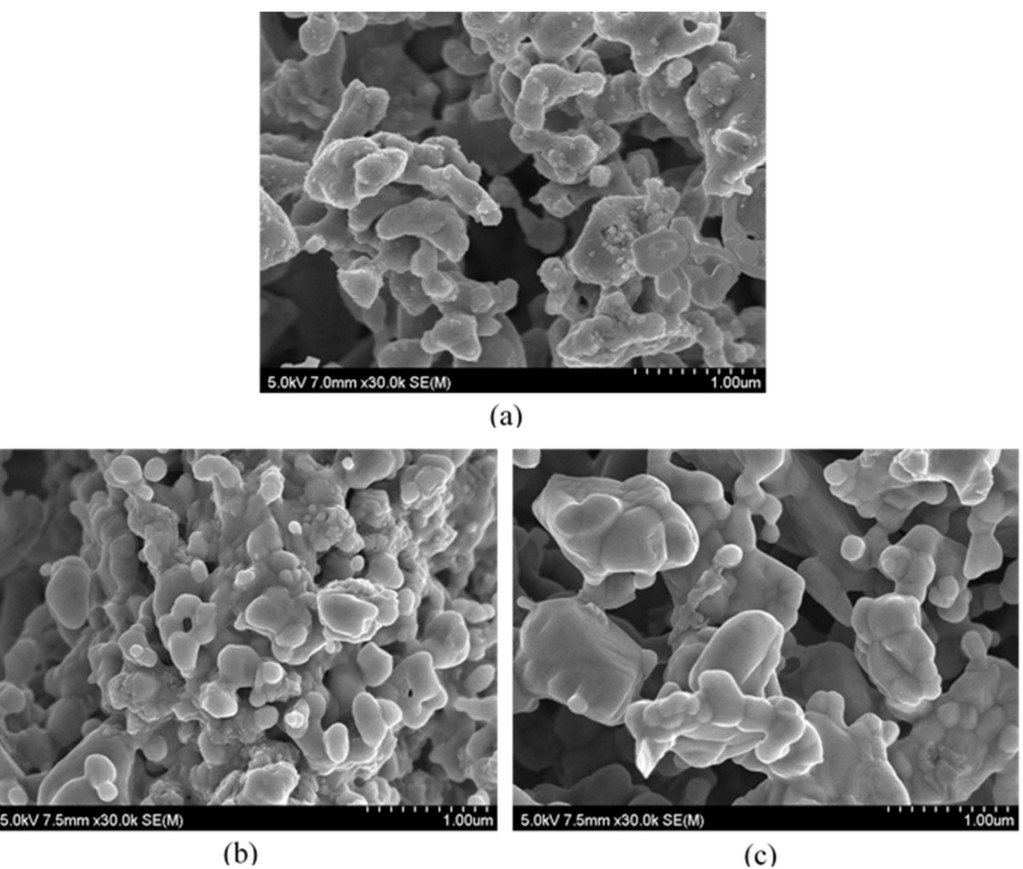

**Figure 6.** SEM images of $SrFeO_{3-\delta}$: (**a**) Before the redox test, (**b**) after six cycles of redox measurements at 650 °C, and (**c**) after five cycles of reproducibility tests at 700 °C.

## 3. Materials and Methods

### 3.1. SrFeO$_{3-\delta}$ Preparation and Characterization

$SrFeO_{3-\delta}$ perovskite-type metal oxide was prepared using solid-state synthesis for use as a $CO_2$ decomposition material. The metal oxide was purchased from K-ceracell Co. Ltd. (Taejeon, Korea), and $SrCO_3$ (Alfa Aesar, >99%, Ward Hill, MA, USA) and $Fe_2O_3$ (Alfa Aesar, >99.9%, Ward Hill, MA, USA) were used as starting materials. After weighing the appropriate amount of materials, the powders

were mixed in ethanol (Samchun Chemicals, >99.9%, Seoul, Korea). The mixed powder was ball-milled with zirconia balls ($\varphi$3–5 mm) for 48 h and then dried to remove the solvent. $SrFeO_{3-\delta}$ powder was heated at 1000 °C for 3 h in ambient air. The final powders were obtained by ball-milling with ethanol for 24 h and then drying in an oven at 80 °C for 24 h.

The synthesized powders were characterized using X-ray powder diffraction (XRD, Rigaku D/MAX-2500, Tokyo, Japan), scanning electron microscopy (SEM, S-4800 Hitachi, Hitachi, Japan), and energy-dispersive X-ray spectroscopy (EDS, Thermo Scientific, Waltham, MA, USA) to analyze the phase structure and purity, surface morphology, and chemical composition, respectively. In-situ XRD (Rigaku D/MAX-2500, Tokyo, Japan) and thermogravimetric analysis (TGA, TA Instruments, Milford, MA, USA) were used to investigate the reduction behaviors of the synthesized powder. In-situ XRD and the change in weight were measured with ≈100 mL $min^{-1}$ of 3.5 vol% $H_2$/Ar.

### 3.2. $CO_2$ Decomposition Experiments

We used a continuous-flow reactor to investigate the products of the $CO_2$ decomposition reaction in the tests. Sample activation and $CO_2$ decomposition were performed with 3.5 vol% $H_2$/$N_2$ and 1 vol% $CO_2$/He, respectively. Approximately 1.5 g of sample powder with zirconia balls ($\varphi$2–3 mm, 10 g) was placed at the center of a quartz tube (I.D.: 12 mm, O.D.: 16 mm, height: 600 mm). The flow rate for the sample activation and $CO_2$ decomposition was 50 mL $min^{-1}$. The $CO_2$ decomposition experiments were performed under increasing temperature at $25 \leq T \leq 800$ °C (nonisothermal). The ramp rate was 3 °C $min^{-1}$. The same measurements were performed at certain fixed temperatures between 500 and 800 °C (isothermal). The residence time was calculated to be 3.393 s. $CO_2$ decomposition cycle tests were also performed to check the regeneration, reproducibility, and stability of the sample. The temperature was selected with the results based on the isothermal $CO_2$ decomposition results. Five cyclic tests under a fixed condition were performed at 700 °C. The blank tests were carried out under identical conditions by using only zirconia balls instead of the sample powder. Table 3 summarizes the experimental conditions for both activation and decomposition. The produced gas concentrations were measured with an Agilent 8890 gas chromatograph (GC, ShinCarbon ST 100/120 micropacked column, Bellefonte, PA, USA). The details of this experiment have been described in a previous report [27].

## 4. Conclusions

We demonstrated that $SrFeO_{3-\delta}$ could serve as a promising material for effective $CO_2$ decomposition in a continuous gas-flow system. In this study, we performed three categorized $CO_2$ treatment experiments using $SrFeO_{3-\delta}$: nonisothermal, isothermal, and stability tests. In nonisothermal experiments, the maximum $CO_2$ decomposition rate reached ~90% and the decomposition rate ($\geq$80%) lasted for around 170 min. Both $CO_2$ decomposition and CO generation by $SrFeO_{3-\delta}$ exhibited better performance than those of $NiFe_2O_{3-\delta}$ or $SrFeCo_{0.5}O_x$. The results of isothermal tests indicated that the optimized temperature for $CO_2$ decomposition should range between 650 and 700 °C. As the operating temperature increased from 625 to 650 °C, the amount of $CO_2$ decomposition increased unusually, even though no special structural phase or weight changes occurred. In addition, $SrFeO_{3-\delta}$ maintained the $CO_2$ decomposition efficiency during isothermal cyclic and reproducibility experiments. Therefore, $SrFeO_{3-\delta}$ is expected to have the capacity to contribute to the mitigation of $CO_2$, a greenhouse gas. Nonetheless, studies of $CO_2$ utilization should be continued because many unsolved problems remain in this field.

**Author Contributions:** Conceptualization, C.Y.P. and S.-C.N.; methodology, J.S.; investigation, S.-H.K. and J.-Y.K.; data curation, C.Y.P. and J.S.; writing—original draft preparation, J.S.; writing—review and editing, K.B.L. and C.Y.P.; supervision, C.Y.P.; project administration, S.-C.N.; funding acquisition, S.-C.N. All authors have read and agreed to the published version of the manuscript.

**Funding:** The authors would like to acknowledge the financial support of the National Research Foundation under the "Next Generation Carbon Upcycling Project" (Project No. 2017M1A2A2043109) of the Ministry of Science and ICT, Republic of Korea.

**Acknowledgments:** The analytical support from the Platform Technology Laboratory at Korea Institute of Energy Research is much appreciated, especially the invaluable assistance provided for in-situ XRD and surface analysis.

**Conflicts of Interest:** The authors declare no conflict of interest.

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
