# Peer review of "Enhanced Carbon Dioxide Decomposition Using Activated SrFeO3−δ"

_catalysts, doi:10.3390/catal10111278_

Round 1
Reviewer 1 Report
In this paper, the authors synthesized SrFeO3-δ and use it for CO2 decomposition. Compared with Ni-ferrite and SrFeCo0.5Ox, SrFeOx has larger CO2 conversion in nonisothermal conditions and great stability under isothermal conditions. In-situ XRD is used to characterize the crystal structure change during the reaction and SEM is used to characterize the morphology change after the reaction. However, there are several issues need to be solved:
- In the abstract, the authors introduced their previous work (SrFeCo0.5Ox for CO2 decomposition, more active than Ni-ferrite). I think it is okay to mention it in the ‘introduction’, but improper to be mentioned in ‘abstract’.
- ‘Chemical decomposition’ is usually described as AB to A + B reactions. However, ‘CO2 decomposition’ described in this paper indicates ‘CO2 + reduced-catalyst to C + oxidized-catalyst’. I suggest it should be considered as ‘CO2 reduction’ or ‘chemical looping system’ (Ref25) if H2 is also included in the reaction. For example, ‘Mg + CO2 to MgO + C’ should be considered as Mg being oxidized by CO2 instead of CO2 decomposition.
- Similar to previous question, I suggest SrFeOx should not be considered as ‘catalyst’. In addition, the decomposed CO2 (3.30 mmol/g) is far less than the ‘catalyst’ if we convert 3.3 mmol CO2 to 145.2 mg CO2.
- Section 2.3 describes how conductivity affect CO2 decomposition effect and explained why they choose SrFeOx for the catalyst. However, the data to prove their suggestion comes from another ref (REF32 and 33) and their own data ‘will be reported in a separate paper’. It is better to add their own data here or move this paragraph to introduction (the reason to choose SrFeOx).
- There is no direct evidence that C is form on the catalyst surface after CO2 decomposition.
- In the introduction, the authors claim that CO2 decomposition method could be used for CO2 storage and utilization and better than CO2 methanization which has high cost and energy. But this CO2 decomposition method also need high temperature and H2 to reduce the catalyst. Additionally, the catalyst should be very sensitive to O2 (and H2O?).
- The CO2 gas used in this work is 1% CO2. Is it possible to use CO2 with high concentration and shorten the reaction time?
Author Response
Dear Editor:
Enclosed is a revised manuscript entitled “Enhanced Carbon Dioxide Decomposition Using SrFeO3-d” by Jaeyong Sim, Sang-Hyeok Kim, Jin-Yong Kim, Ki Bong Lee, Sung-Chan Nam, and Chan Young Park. This revised manuscript incorporates the suggestions made by the reviewers on October 13, 2020.
We have provided our responses to the reviewer’s comments, including details of the additions and revisions made to the manuscript, in the enclosed “Response to reviewers” letter. All references, units, and figure captions have been prepared in keeping with the journal’s Guide for Authors.
Please feel free to contact me if you have any questions regarding the manuscript. Thank you for your time and consideration.
Best regards,
Chan Young Park

Reviewer 2 Report
This manuscript deals with the decomposition of CO2 to CO using SrFeO3 as a catalyst. SrFeO3 was first activated – deprived of oxygen, in a H2 environment followed by exposure to CO2 where the oxygen vacancy created during H2 exposure was refilled with oxygen from CO2 and CO2 is reduced to CO. The claims are well supported by the results. My concern regarding this work is,
- How is this a catalytic process if it constitutes the reaction of CO2 with the vacancy sites of the electrode until the electrode oxygen vacancy is completely consumed.
- For the activation and cycling purposes, the electrode was exposed to hydrogen and the decomposition of CO2 to CO happens at high temperatures like 600 – 800C. At these temperatures, hydrogen can readily react with CO2 to decompose it to CO and water. In comparison to that how efficient or economical this process is not clear.
- What is the stoichiometric concentration of oxygen in the SrFeO3-d and how much of oxygen is actually removed during activation needs to be clearly informed.
- Is the higher performance of SrFeO3 in comparison to SrFeCo0.5O3-d and NiFe2O3-d is due to a higher inherent oxygen vacancy? This needs to be discussed.
- At lower temperature operations like 700C, considering the slow removal and refilling of oxygen vacancies, why the CO production did not extend beyond the 150 minutes of operation (Figure 3b).
Author Response

(The authors gave the same response as above.)

Reviewer 3 Report
The article discusses CO2 decomposition in the presence of SrFeO3-x catalyst in a continuous flow system. An enhanced catalytic efficiency and stability were reported. CO2 decomposition was performed under isothermal and non-isothermal experimental conditions. A maximum decomposition of 90% was reported and the activity last for close to 3 hours.
Following revisions are recommended:
1) Addition of description of employing H2 and forming CO as product is recommended in the abstract.
2) Based on the reported results, it looks like it is a reverse water gas shift reaction. How economical is this process? Hydrogen is expensive. The use of hydrogen for the decomposition should be discussed in the introduction section.
3) What are the dotted lines in Figure 2a and 2b; description for the dotted lines are missing and thus the figures are not discernible.
4) Figure 3 and 4, include the blank experimental conditions.
Author Response

(The authors gave the same response as above.)

Round 2
Reviewer 1 Report
Thanks for the explanation. All of my concerns have been mentioned in the response. I do not have any other questions.